# First Isolation of *Pseudogymnoascus* *destructans*, the Fungal Causative Agent of White-Nose Syndrome, in Korean Bats (*Myotis petax*)

**DOI:** 10.3390/jof8101072

**Published:** 2022-10-12

**Authors:** Young-Sik Kim, Myun-Sik Yang, Chang-Gi Jeong, Chul-Un Chung, Jae-Ku Oem

**Affiliations:** 1Department of Veterinary Infectious Disease, College of Veterinary Medicine, Jeonbuk National University, Iksan 54596, Korea; 2Department of Life Science, Dongguk University, Gyeongju 04620, Korea

**Keywords:** white-nose syndrome, *Pseudogymnoascus* *destructans*, Korean bats, *Myotis* *petax*, first isolation

## Abstract

White-nose syndrome (WNS), caused by *Pseudogymnoascus* *destructans* (*Pd*), is a lethal fungal disease that affects hibernating bats in North America. Recently, the presence of *Pd* was reported in countries neighboring Korea. However, *Pd* has not been investigated in Korea. Therefore, this study aimed to identify the presence of *Pd* in Korean bats. Altogether, wings from 241 bats were collected from 13 cities and cultured. A total of 79 fungal colonies were isolated, and two isolates were identified as *Pd* using polymerase chain reaction. Of the nine bat species captured in 13 cities, *Pd* was isolated only from *Myotis* *petax* in Goryeong. Atypical, curved conidia were observed in two isolated fungal colonies. Although histological lesions were not observed by hematoxylin and eosin or periodic acid–Schiff staining, fungal invasion was observed in the tissue sections. Taken together, these results confirmed the presence of *Pd* in Korean bats and suggest the possibility of WNS outbreaks in Korean bats. This is the first report of the isolation and molecular analysis of *Pd* from Korean bats.

## 1. Introduction

The emergence of infectious diseases reduces the population of wild animals and adversely affects ecosystems [1]. Various pathogens have been reported worldwide [2]. Therefore, it is necessary to investigate the causative agents and distribution of infectious diseases. By understanding the distribution of pathogens, measures can be taken to reduce the risk of disease spread [1].

White-nose syndrome (WNS) is characterized by the growth of the white fungus *Pseudogymnoascus destructans* (*Pd*) on the muzzles, ears, and wings of infected bats [3,4,5]. Fungi around the muzzle represent the most dramatic sign of WNS infection. However, the most important target of *Pd* is the skin of bat wings, which plays a critical role in homeostasis [6,7,8]. In contrast to the typical infectious dermatophytes of mammals that can only invade inanimate skin structures (keratin, hair, and nails), *Pd* can erode and invade extensive areas by digesting the skin and underlying connective tissue of hibernating bats [6,7]. Histologically, a characteristic cup-like lesion is observed after WNS infection. In infected bats, ulcers and fungal invasion of these tissues are commonly observed, but no inflammation has been noted in response to fungal hyphae [6]. In addition, no lesions have been found in tissues other than the skin. Recovery after infection may involve fungal spores surrounded by a thin layer of acellular material in the dermis or epidermis [6].

*Pd* is a zoophilic fungal pathogen that causes WNS in hibernating bats [3,4,5]. Since first being confirmed in 2006, WNS has been reported in 38 states in the United States and seven provinces of Canada [9]. Subsequently, *Pd* has been reported in 23 countries across Europe and in five Asian countries [10]. WNS has killed more than 6.7 million bats in the Nearctic region, but no mass deaths due to this disease have been reported in the Palearctic region [3,4,9,10]. More than 1300 bat species are known to exist worldwide, and 12 and 14 WNS-affected species have been reported among bat species in the Nearctic and Palearctic regions, respectively [9].

An outbreak of WNS in Korea can lead to serious consequences for some bat species. Among bats living in Korea, 14 species have been reported to be affected by WNS in other countries [9,11]. In the present study, two *Pd* species were isolated from bat wing tissues and characterized using molecular analysis and histological assays.

## 2. Materials and Methods

### 2.1. Sample Collection

All bat samples were obtained from the National Institute of Wildlife Disease Control and Prevention through the appropriate procedures. Altogether, 241 bats were captured in 13 cities belonging to six provinces of South Korea from August to November 2021 (Figure 1). All bats used in this study were captured before hibernation, and 20 bats were captured from 11 cities each, except for Gunwi (2) and Pocheon (19). Bats were captured from caves and forests using a bat mist net. Experts studying bat ecology confirmed the species and sex of the captured bats. Information regarding the captured bats is presented in Appendix A. All bats were examined for the presence of *Pd* using ultraviolet (UV) light with a wavelength of 365 nm (Analytik, Jena, Germany), as described in a previous study [12]. All samples were transported to the laboratory in a cooler.

### 2.2. Culture and Isolation of Fungi

A total of 241 bat wing membranes were cultured on Sabouraud dextrose agar (SDA) and dextrose peptone yeast extract agar (DPYA) containing 100 μg/mL gentamycin and chloramphenicol (MBcell, Seoul, Korea). Bat wings were wiped ten times on both sides with 70%-alcohol-containing cotton to minimize contamination of previously buried fungal spores. Bat wing membranes were selected for thinner or nodular areas because there was no fluorescence response to UV light. The wing membranes were cut into 2 × 2 cm sizes and placed on the media to be cultured. Media planted with bat wing membranes were dark-incubated at low temperatures (6 °C) for 40 days [13]. The media were checked daily. The single-spore isolation method was used to obtain pure fungal cultures after collecting all colonies from the primary culture media [14]. Single fungal cells were washed once with distilled water and harvested in 2 mL phosphate-buffered saline. The isolated *Pd* strains were deposited in the Korea Veterinary Culture Collection.

### 2.3. Genomic DNA Extraction and Molecular Identification

The total DNA of single fungal colonies was extracted using the i-genomic BYF DNA Extraction Mini Kit (iNtron Biotechnology, Seongnam, Korea) according to the manufacturer’s instructions. Genomic DNA was used for the amplification and sequencing of five markers using the modified Minnis and Linder’s method [15]: ribosomal internal transcribed spacer (ITS), 28S large subunit (LSU) rRNA gene, translation elongation factor 1 alpha (TEF1-α), RNA polymerase II second-largest subunit (RPB2), and minichromosomal maintenance protein 7 (MCM7). The primer sequences used to amplify the markers are listed in Table 1. Amplification was performed using Maxime PCR Premix i-StarTaq (iNtron Biotechnology, Seongnam, Korea). The total polymerase chain reaction (PCR) mixture (20 µL) contained 50 ng of DNA template and 10 pmol of each primer. The amplified fragments were purified using a LaboPass PCR Purification Kit (CosmoGeneTech, Inc., Seoul, Korea). After purification, all the samples were sequenced using barcode-tagged sequencing using a commercial sequencing service (BTSeqTM; CELEMICS, Seoul, South Korea). All sequences generated in this study have been deposited in the GenBank database, and their accession numbers are listed in Appendix A.

### 2.4. Molecular and Phylogenetic Analysis

The fungal sequences obtained were identified using a BLAST search “http://www.ncbi.nl-m.nih.gov“ (accessed on 26 January 2022). Available reference sequences were retrieved from the NCBI database, and fungal sequences were aligned with the reference sequences using MAFFT v7.3113. Sequence editing and concatenation were performed using MEGA 7.0.26 [16]. Minnis and Lindner observed that LSU and TEF1 introns have limited phylogenetic value because they are present and scattered among unrelated members of *Pseudogymnoascus* [15]. Therefore, homologous gaps corresponding to LSU and TEF1 introns were excluded. In addition, the non-overlapping ends of the sequences in each alignment were trimmed. Phylogenetic analysis was conducted using maximum likelihood (ML) and Bayesian inference (BI) methods. ML analyses were performed using IQ-TREE v 1.6.8 [17]. The best fit nucleotide substitution model for each locus was estimated using the IQ-TREE model finder function [18] following the Bayesian information criterion (BIC). Bootstrap analyses were performed using ultrafast bootstrap approximation with 1000 replicates [19]. BI analyses were performed using MrBayes v.3.2.6 [20]. The analyses included two independent runs of 5 million generations with four chains each. The substitution model was set to K2 + I + G, and the first 25% of the samples and trees were discarded as burn-ins. In addition, phylogenetic tree analysis was performed using the ML method for each of the five gene loci, and the differences in genes between *Pd* in other regions were compared and analyzed.

### 2.5. Histopathological Examination

The wing tissues of the bats were immediately fixed in 10% formalin, dehydrated, and embedded in paraffin. To examine the histopathological findings, sections were prepared from each wing membrane biopsy specimen and stained with periodic acid–Schiff (PAS) and hematoxylin and eosin (H&E). Fungal conidia isolated from the media were stained with PAS and observed morphologically.

## 3. Results

### 3.1. Isolation of Fungi

All captured bats were negative for fluorescence reactions under UV light. Forty fungal colonies were obtained from the DPYA medium, and 39 fungal colonies were collected from the SDA medium.

A total of 79 isolates were obtained from 241 bat wing tissue samples and subjected to ITS rRNA sequencing. Phylogenetic analysis classified these isolates into the phylum Ascomycota, which corresponds to 28 genera in 10 orders (Figure 2). The isolated fungal orders included Microascales (22.8%), Pleosporales (21.5%), Incertae sedis (19.0%), Capnodiales (10.1%), and Eurotiales (10.1%) (Figure 2). A variety of other fungal genera were isolated from bat wing tissues, including, in order of relative abundance, *Arthrinium* (16.5%), *Kernia* (13.9%), *Alternaria* (10.1%), *Penicillium* (10.1%), *Cladosporium* (8.9%), and *Microascus* (5.1%). All these were considered incidental findings. In addition, two *Pseudogymnoascus* (2.5%) isolates were confirmed, and molecular genetic analysis was performed (Figure 2). Of the nine bat species captured in 13 cities, these two *Pseudogymnoascus* strains were isolated only from *Myotis petax* in Goryeong.

### 3.2. Molecular and Phylogenetic Analysis

The ITS gene sequences of the two *Pseudogymnoascus* isolates (BW48 and BW49) were identified through a BLAST search, and the sequences matched 100% with the North American *Pd* isolates (NR_111838) and European isolates (KF866377, GQ489024, MK421359, LN871244). However, a 99.79% match was observed with the European (KF866378, LN871246), North American (KF212292, KF212293), and Russian (LN871280, LN871291) isolates (Figure 3). LSU sequences showed 99.70%–99.71% similarity with North American *Pd* isolates and 99.7% similarity with European British isolates (Figure 3). For MCM7 gene sequences, 99.8% of the North American *Pd* isolates were consistent (Figure 3). The TEF1 sequences were 100% consistent with the European *Pd* isolates. In contrast, the North American (KJ938428, KJ938429, and KF686768) and some European *Pd* isolates (LN871360, LR736723, and LN871383) (Figure 3) were 99.88% matched. The RPB2 region sequence could not be analyzed because of the lack of a *Pd* reference sequence. In sequence analysis of five gene marker concatenations, two isolates showed no genetic differences and were 99.87% similar to the *Pd*-type strain.

Phylogenetic analyses based on five gene loci showed that two isolated *Pseudogymnoascus* strains were located in the F clade, which included a *Pd*-type strain (Figure 4). Clades were identified using clade nomenclature (A to L), formally defined by Minnis and Lindner [15].

### 3.3. Morphological Characteristics of Pd

The morphology of the fungus identified as *Pd* was examined. *Pd* was secondarily cultured using a single-spore isolation method and changed to a gray color over time in white cottony fungal colonies. There was no pigmentation on the posterior side; however, a brown pigment was secreted from the fungal colony cultivated on SDA medium after further growth (Figure 5).

### 3.4. Histopathological Findings

PAS and H&E staining did not reveal the characteristic cup-like lesions caused by WNS in paraffin sections of the tissue observed under a microscope. However, fungal spores surrounded by acellular material were identified in the epidermis and dermis of the bat wings (Figure 6a,b). Microscopy of the isolated *Pd* revealed distinct curved fungal conidia (Figure 6c).

## 4. Discussion

During hibernation, the body temperature of bats drops close to ambient temperatures (usually 2–10 °C), and simultaneously, their metabolic rate decreases by 96–98% [21,22]. Studies have shown that hibernation in mammals downregulates the immune response, which does not return to normal reaction until the basal metabolic rate and core temperature return to euthermic levels [21]. Among the 241 bats of nine species caught in 13 cities in Korea, *Pd* was isolated from only 2 of 38 *Myotis petax* that were captured in Goryeong. The Goryeong region, where *Myotis petax* was captured, is located adjacent to the Nakdong River, with a stream flowing in the center and a nearby reservoir. This region is surrounded by mountains and has conditions suitable for *Myotis petax* survival. *Myotis petax* is a species closely related to the water system and is active at night in lakes, rivers, and ponds [11]. *Myotis petax* uses tree holes, caves, or artificial structures such as bridges as a refuge in summer and hibernates alone or in groups of several dozens in winter [11]. Moreover, *Myotis petax* have a habit of nesting themselves in the crevices of rocks and hibernating [22]. These habits increase the probability of *Pd* infection by increasing the frequency of contact with *Pd*-contaminated cave environments or *Pd*-infected bats. In addition, it has been reported that *Myotis petax* prefers a lower temperature of the rock wall (4.4 ± 1.1 ℃) when selecting hibernacula compared to other species [22]. This behavior provides an appropriate environment for the growth of *Pd* on the bat skin. Based on the habits of these bats, of the nine species of bats in this study, *Myotis petax* is most likely to be infected with *Pd*.

Bats play an important role in the fungal diversity of caves [23]. *Arthrinium*, *Kernia*, *Alternaria*, *Penicillium*, *Cladosporium*, and *Microascus*, which are relatively abundant among the 20 fungal genera cultured in this study, have been reported in previous studies of cave fungi and are closely related to cave environments [24,25,26]. We infer that the attachment of fungal spores to the skin or fur of bats occurred while they inhabited cave environments.

Fungal cultures and PCR are useful diagnostic methods for *Pd* [27]. Similar methods were used to isolate and genetically analyze the isolates in this study. The results of the analysis of concatenating the five gene loci of isolated *Pd* in this study showed 99.88% similarity to the *Pd*-type strain, and the genetic difference may be a result of the genotypic differences between North America and East Asia [10,28].

Histopathology is considered an effective method for diagnosing WNS [27]. Clinical signs such as powdery white spores in the infected regions and cup-like lesions unique to *Pd* infection have been commonly observed [3,6]. However, since this study started in late summer (30 August; 22–25 °C) before hibernation, we could not confirm these characteristic clinical signs or histopathological evidence. Nevertheless, fungal spores surrounded by a thin layer of acellular material observed in bats infected with WNS have been identified in the dermal and epidermal layers, as confirmed in a previous study [6]. The results of this study suggest that bats were infected with WNS during hibernation and recovered after awakening.

Based on molecular, biological, and histological evidence, *Pd* has been demonstrated to exist in Korea. In the current study, *Pd* was first isolated from bats living in Korea, although the outbreak of WNS was not confirmed for various reasons, such as the season in which bats were captured. Nevertheless, an outbreak of WNS is possible in the near future; therefore, continuous investigation and research are needed.

## Figures and Tables

**Figure 1 jof-08-01072-f001:**
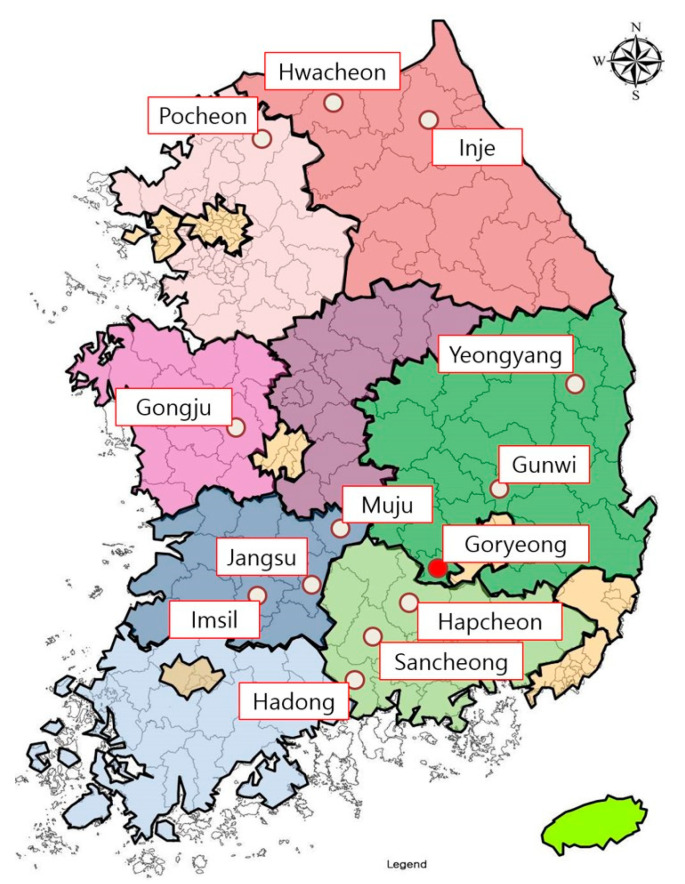
Bat capture sites in this study. Altogether, 241 bats were captured in 13 cities belonging to six provinces of South Korea from August to November 2021. The city in which bats with *Pd* were captured is marked with a red filled circle.

**Figure 2 jof-08-01072-f002:**
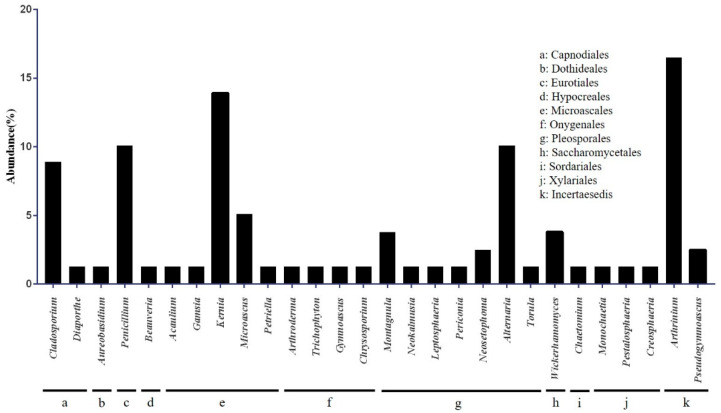
Classification of cultured fungi at the class and genus levels. Seventy-nine fungal colonies were isolated from the two media. *Microascaceae* was the most dominant family, followed by *Apioporaceae* and *Pleosporaceae*.

**Figure 3 jof-08-01072-f003:**
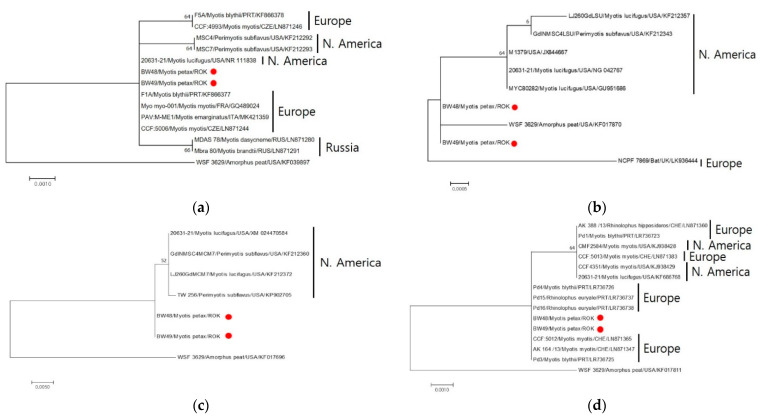
Maximum likelihood tree based on the DNA sequences of four loci. Comparative analysis of *Pseudogymnoascus destructans* (*Pd*) was based on separate analyses of four loci: ITS (**a**), LSU (**b**), MCM7 (**c**), and TEF1-a (**d**). The tree was built using MEGA 7.0.26 [16] with the best fit model. Branch support values were obtained using 1000 bootstrap replicates.

**Figure 4 jof-08-01072-f004:**
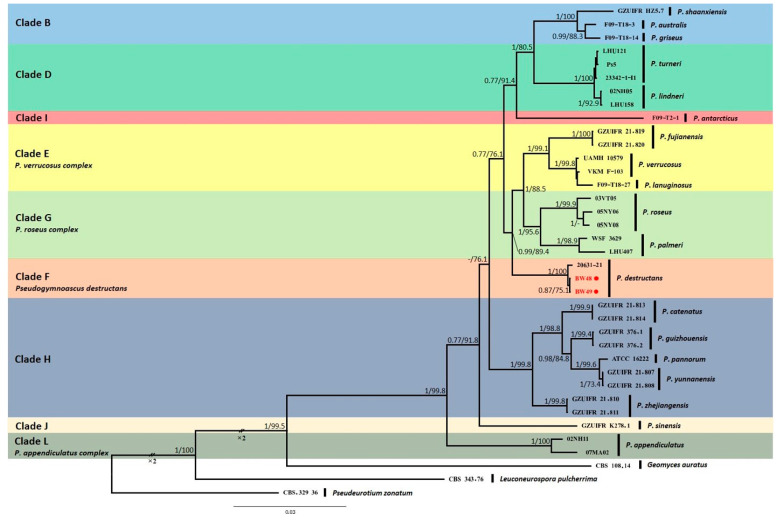
Bayesian inference phylogenetic tree of *Pseudogymnoascus* genus generated from the concatenated dataset of five loci (ITS, LSU, TEF1, RPB2, and MCM7). Bayesian posterior probabilities (BPPs) and significant maximum likelihood bootstrap (BS) values are indicated by branches. Only clades that received 0.75 BPPs and 70% BS simultaneously were considered strongly supported and presented at the branches. Clades were identified using clade nomenclature (A to M), formally defined by Minnis and Lindner [15]. The scale bar indicates 0.03 nucleotide changes per site. *Pd* is highlighted in red and with a red spot.

**Figure 5 jof-08-01072-f005:**
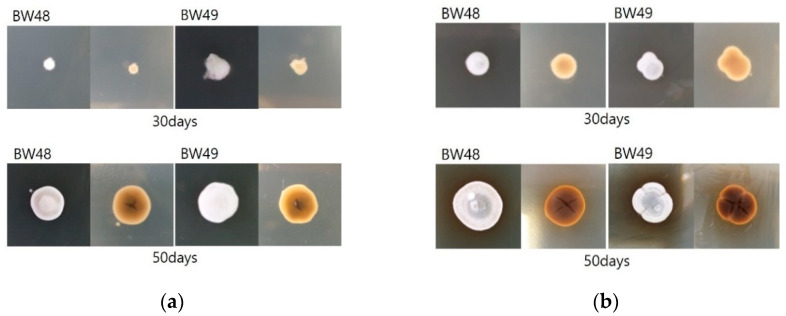
Morphology of isolated *Pd* according to the time period for growth. Cultured fungal colonies changed from a white and cottony appearance to gray, and fungal colonies cultured on SDA medium secreted a brown pigment: (**a**) DPYA Media; (**b**) SDA Media.

**Figure 6 jof-08-01072-f006:**
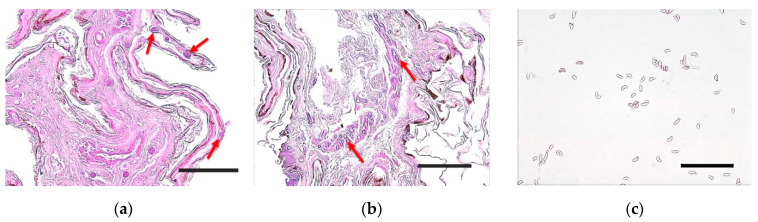
Microscopic analysis of tissue sections and fungal conidia. Hematoxylin and eosin (**a**) and periodic acid–Schiff (**b**) staining of the bat wing tissue sections. Fungal spores were surrounded by a thin layer of acellular material in the skin tissue of the bats (red arrows) (bar = 50 μm). (**c**) Periodic acid–Schiff-stained conidia of isolated *Pd*. Fungal conidia of the distinctive curved forms of *Pd* were identified (bar = 20 μm).

**Table 1 jof-08-01072-t001:** Primer information used in this study.

Gene Marker	Primer	Sequence
ITS ^1^	ITS1	TCCGTAGGTGAACCT GCG
	ITS4	TCCTCCGCT TATTGATATGC
LSU ^2^	LR0 R	ACCCGCTGAACTTAAGC
	LR7	TACTACCACCAAGATCT
Tef1-α ^3^	EF1-983 F	GCYCCYGGHCAYCGTGAYTTYAT
	EF1-2218 R	ATGACACCRACRGCRACRGTYTG
MCM7 ^4^	MCM7-709 for	ACIMGIGTITCVGAYGTHAARCC
	MCM7-1348 rev	GAYTTDGCIACICCIGGRTCWCCCAT
RPB2 ^5^	fRPB2-7 cF	ATGGGYAARCAAGCYATGGG
	RPB2-3053 bR	TGRATYTTRTCRTCSACCAT

^1^ ITS: internal transcribed spacer, ^2^ LSU: large subunit, ^3^ Tef1-α: translation elongation factor 1-alpha, ^4^ MCM7: minichromosome maintenance complex component 7, ^5^ RPB2: RNA polymerase II second-largest subunit.

## Data Availability

Isolated fungal sequences were submitted to the NCBI GenBank database under accession numbers.

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
