# Peer review of "First Isolation of Pseudogymnoascus destructans, the Fungal Causative Agent of White-Nose Syndrome, in Korean Bats (Myotis petax)"

_jof, 2022, doi:10.3390/jof8101072_

Round 1

Reviewer 1 Report

This manuscript offers a well-organized summary of the authors' highly significant determination that Pseudogymnoascus destructans (Pd) is present, although in two of 241 captured Myotis petax captured in 13 cities distributed among six provinces, in Korean bats. The authors research contributes important insight into ongoing challenges to identify and characterize worldwide geographic habitats and hosts of Pd, as the global community continues to strategize to control the spread of white nose syndrome.

The following suggestions are for your consideration:

Line 6: Capitalize Veterinary Medicine

Line 13-4: reported in countries neighboring Korea.

Line 15-16: wings from 241 bats were collected    (as in line 58)

Line 23:  Korean bats

Line 147:  Change Microasccaceae to Microascaceae

Line 179: omit comma (perhaps/consider)

Line 92, 198, 199, 206, 285: Italicize Pd or Pseudogymnoascus destructans, Geomyces destructans

Line 237: Consider revising 'treated with WNS' to (perhaps) manifesting WNS or displaying WNS

Line 282: omit the word journal (as in line 285)

Author Response

Reviewer #1:

Comments and Suggestions for Authors

This manuscript offers a well-organized summary of the authors' highly significant determination that Pseudogymnoascus destructans (Pd) is present, although in two of 241 captured Myotis petax captured in 13 cities distributed among six provinces, in Korean bats. The authors research contributes important insight into ongoing challenges to identify and characterize worldwide geographic habitats and hosts of Pd, as the global community continues to strategize to control the spread of white nose syndrome.

The following suggestions are for your consideration:

Line 6: Capitalize Veterinary Medicine

RESPONSE: We corrected the sentences in line 6.

Line 13-4: reported in countries neighboring Korea.

RESPONSE: We revised this sentences from ‘reported in neighboring countries in Korea’ to ‘reported in countries neighboring Korea’ in lines 13-14.

Line 15-16: wings from 241 bats were collected (as in line 58)

RESPONSE: We revised this sentence from ‘241 wings of bat were collected and cultured’ to ‘wings from 241 bats were collected from 13 cities and cultured’ in lines 15-16.

Line 23:  Korean bats

RESPONSE: We corrected the words from ‘korean bats’ to ‘Korean bats’ in line 24.

Line 147:  Change Microasccaceae to Microascaceae

RESPONSE: We corrected the words from 'Microasccaceae' to 'Microascaceae' in line 151.

Line 179: omit comma (perhaps/consider)

RESPONSE: We deleted comma in this sentence.

Line 92, 198, 199, 206, 285: Italicize Pd or Pseudogymnoascus destructans, Geomyces destructans

RESPONSE: We corrected the Latin names of the entire sentence to italic.

Line 237: Consider revising 'treated with WNS' to (perhaps) manifesting WNS or displaying WNS

RESPONSE: We revised this sentences from ‘treated with WNS’ to ‘infected with WNS’ in line 241.

Line 282: omit the word journal (as in line 285)

RESPONSE: We deleted word ‘journal’ in reference 4.

Reviewer 2 Report

Comments and suggestions for authors see in an enclosed pdf file.

Author Response

Reviewer #2:

 GENERAL COMMENTS

In the manuscript, new data on P. destructans are given. I have only some minor comments.

Introduction

  • It is not emphasized in the introduction that while in the US the fungus attacked a naive population of bats and caused massive deaths, this situation did not occur in other areas.

RESPONSE: We added the sentence ‘WNS has killed more than 6.7 million bats in Nearctic region, but no mass deaths due to this disease have been reported in the Palearctic region’ in lines 49-51.

OTHER COMMENTS

Abstract

  • Instead of 241 wings sampled it could be given the number of bats (241), and number of sites (13).

RESPONSE: We revised this sentence from ‘241 wings of bat were collected and cultured’ to ‘wings from 241 bats were collected from 13 cities and cultured’ in lines 15-16.

  • Regarding the two Pd isolates, it should be noted that they come from one locality.

RESPONSE: We added a sentence that ‘Of the nine bat species captured in 13 cities, Pd was isolated only from Myotis petax in Goryeong.’ in lines 17-18.

Material and Methods

  • Some details on sampling could be given. How many bats were sampled from each site (give at least a range, e.g. 5-8)?

RESPONSE: We added a sentence that ‘All bats used in this study were captured before hibernation, and 20 bats were captured from 11 cities each, except for Gunwi (2) and Pocheon (19).’ in lines 63-64.

  • Were the bats killed after sampling? Does the responsible authors have permission for such work?

RESPONSE: The bats used in this experiment were autopsied to be used for other experiments, and we conducted the experiment by receiving unused bat wings. The supply of bat wings was made at the site of an autopsy.

  • r. 63: …as described previously study… x …as described in a previous study…

RESPONSE: We revised this sentence from ‘as described previously study’ to ‘as described in a previous study’ in line 68.

  • r. 80: explain the abbreviation PBS

RESPONSE: We revised this word from ‘PBS’ to ‘phosphate-buffered saline’ in line 85.

  • It could be emphasized that the bats were captured before hibernation.

RESPONSE: We added a sentence that ‘All bats used in the experiment were captured before hibernation, and 20 were captured in 11 cities each except Gunwi (2) and Pocheon (19).’ in lines 62-64.

Results

  • Delete the rows 128-130.

RESPONSE: We deleted these sentences.

  • R. 132-133: I don't understand the sentence. In the first part it is stated that the UV reaction was confirmed, but in the second that it was negative.

RESPONSE: We revised ‘All captured bats confirmed fluorescence reactions using UV lamps; however, all were UV-negative.’ to ‘All captured bats were negative for fluorescence reactions under UV-ray.’ to clarify the meaning of this sentence in line 135.

  • Fig. 2, r. 147: Microascaceae instead of Microasccaceae

RESPONSE: We corrected the words from 'Microasccaceae' to 'Microascaceae' in line 151.

  • Fig. 2, r. 148: Pleosporaceae instead of Peleosporaceae

RESPONSE: We corrected the words from 'Peleosporaceae' to ' Pleosporaceae' in line 152.

  • R. 150: I consider it appropriate to add the numbers of the isolates (BW48, and BW49).

RESPONSE: We added the number of the isolates in line 154.

  • The P. destructans isolates should be deposited in a publicly available fungal culture collection.

RESPONSE: We deposited that isolated Pd in the Korea Veterinary Culture Collection. And we described that in lines 85-86.

  • R. 158: North American instead of North America

RESPONSE: We corrected this word from ‘North America’ to ‘North American’ in line 163.

  • R. 164: Pseudodymnoascus strains instead of Pseudogymnoascus genera

RESPONSE: We revised this word from ‘genera’ to ‘strains’ in line 169.

  • Fig. 3b: Myotis petax instead of Myotis Petax

RESPONSE: We corrected Fig 3b from ‘Myotis Petax’ to ‘Myotis petax

  • Fig. 3, 4, 5, 6: Pseudogymnoascus destructans instead of Pseudogymnoascus destructans (italics)

RESPONSE: We corrected the Latin names of the entire sentence to italic.

  • Fig. 4: Pseudeurotium instead of Pseudoeurotium

RESPONSE: We corrected Fig 4 from ‘Pseudoeurotium’ to ‘Pseudeurotium’

  • R. 188: Histopathological instead of Histopathologic

RESPONSE: We corrected this word from ‘Histopathologic’ to ‘Histopathological’ in line 193.

  • Fig. 6c: Conidia should be photographed at higher magnification (1000x) to better see their shape.

RESPONSE: We tried to photograph the fungal conidia at a higher magnification. However, the highest magnification of the microscope available was 600x.
The magnification of the previous picture was 400x, and we revised Figure 6c with 600x.

We described this in Figure Legend of Fig 6c.

Discussion

  • R. 205-206: The data from sentence “Among the 241 bats of 9 species caught in 13 cities in Korea, Pd was isolated only 2 of 38 Myotis petax that were captured in Goreong” should be emphasized in Results and Abstract.

RESPONSE: We added a sentence that ‘Of the nine bat species captured in 13 cities, Pd was isolated only from Myotis petax in Goryeong.’ in lines 17-18. In addition, we added a sentence that ‘Of the nine bat species captured in 13 cities, these two Pseudogymnoascus strains were isolated only from Myotis petax in Goryeong.’ in lines 147-148.

Supplementary materials

  • R. 247: Delete “TableS1: title”

RESPONSE: We deleted these words

References

  • References should be given according to instructions for authors. For example, use lowercase letters in article title words, use italics for Latin names

RESPONSE: We revised the article title word to lowercase and Latin names to italics according to the author's instructions for reference.

Table S2

  • Species names should be listed alphabetically.

RESPONSE: The bat species name shown in Table s2 was listed in conjunction with the sample number. If listed alphabetically, the association with the sample number is broken.